# Gait analysis: An effective tool to mechanically monitor the bone regeneration of critical-sized defects in tissue engineering applications

**Pablo Blázquez-Carmona**[1,2]\*, **Juan Mora-Macías**[2,3], **Juan Morgaz**[4], **María del Mar Granados**[4], **Jaime Domínguez**[1,2], **Esther Reina-Romo**[1,2]

1 Department of Mechanical and Manufacturing Engineering, Escuela Técnica Superior de Ingeniería, Universidad de Sevilla, Seville, Spain, 2 Instituto de Biomedicina de Sevilla (IBiS), University of Seville, Seville, Spain, 3 Department of Mining, Mechanical, Energy and Building Engineering, Escuela Técnica Superior de Ingeniería, University of Huelva, Huelva, Spain, 4 Department of Animal Medicine and Surgery, Universidad de Córdoba, Campus Universitario de Rabanales, Córdoba, Spain

\* pbcarmona@us.es

**Data Availability Statement:** All relevant data are within the paper and its Supporting Information files.

## Abstract

### Introduction

Tissue engineering has emerged as an innovative approach to treat critical-size bone defects using biocompatible scaffolds, thus avoiding complex distraction surgeries or limited stock grafts. Continuous regeneration monitoring is essential in critical-size cases due to the frequent appearance of non-unions. This work evaluates the potential clinical use of gait analysis for the mechanical assessment of a tissue engineering regeneration as an alternative to the traditional and hardly conclusive manual or radiological follow-up.

### Materials and methods

The 15-mm metatarsal fragment of eight female merino sheep was surgically replaced by a bioceramic scaffold stabilized with an external fixator. Gait tests were performed weekly by making the sheep walk on an instrumented gangway. The evolution of different kinematic and dynamic parameters was analyzed for all the animal's limbs, as well as asymmetries between limbs. Finally, potential correlation in the recovery of the gait parameters was evaluated through the linear regression models.

### Results

After surgery, the operated limb has an altered way of carrying body weight while walking. Its loading capacity was significantly reduced as the stance phases were shorter and less impulsive. The non-operated limbs compensated for this mobility deficit. All parameters were normalizing during the consolidation phase while the bone callus was simultaneously mineralizing. The results also showed high levels of asymmetry between the operated limb and its contralateral, which exceeded 150% when analyzing the impulse after surgery. Gait recovery significantly correlated between symmetrical limbs.

**Funding:** This project was supported by the Ministerio de Ciencia e Innovación (Government of Spain) through the grant number PID2020-113790RB-I00 awarded to JMM and ERR, https://www.aei.gob.es/convocatorias/buscador-convocatorias/proyectos-idi-2020-modalidades-retos-investigacion-generacion, and by the Junta de Andalucía (FEDER-UHU, Programa Operativo FEDER de Andalucía 2014-2020) through the grant number UHU-202058 awarded to JMM, https://www.agenciaandaluzadelaenergia.es/es/financiacion/incentivos-2017-2020/fondo-feder-2014-2020. The funders had no role in study design, data collection and analysis, decision to publish, or preparation of the manuscript" was included in the cover letter of the reviewed manuscript.

**Competing interests:** The authors have declared that no competing interests exist.

## Conclusions

Gait analysis was presented as an effective, low-cost tool capable of mechanically predicting the regeneration of critical-size defects treated by tissue engineering, as comparing regeneration processes or novel scaffolds. Despite the progressive normalization as the callus mineralized, the bearing capacity reduction and the asymmetry of the operated limb were more significant than in other orthopedic alternatives.

## Introduction

Bone is a highly dynamic hierarchical composite with homeostatic mechanisms in charge of its continuous remodeling and regeneration in the event of defects caused by tumors, traumas, or infections. Nevertheless, the regenerative capacity of hard tissue is not unlimited. These defects have a size upper limit from which this rigid organ does not heal spontaneously, usually leading to non-unions [1]. Thus, the treatment of critical-sized bone defects is still today one of the significant challenges for acute care surgeons. To address this issue, they must resort to other clinical strategies, including those based on the Ilizarov method (distraction osteogenesis) or the incorporation of substitutes. For instance, bone transport trusts on the long-standing distraction osteogenesis to gradually move an osteotomized surrounded bony fragment towards the position of the original defect while simultaneously forming a bone callus on the other side, the docking site [2–4]. However, this regeneration process carries inherent risks, particularly concerning viscoelastic and structural alterations in the neighboring soft tissues (e.g., skin, tendons, or muscles) [5–8]. Another traditional solution lies in incorporating autogenic and allogenic bone grafts to encourage cell regenerative activity in the gap, mainly vascularized free fibular or Papineau open cancellous bone grafting [9–11]. Despite being the most suitable substitute concerning biocompatibility, osteoinductive and osteoconductive properties, the availability of harvested autografting tissue is limited, and the allograft's growing clinical demand exceeds the donor stock [12,13]. Since not long ago, research for novel alternatives to tissue transfer has been ongoing through tissue engineering (TE) [14]. TE is focused on finding the standard substitute material and structure to effectively emulate the intricate microenvironment of native bone tissue and further promote regeneration [15,16]. In this field, additive manufacturing techniques have gained popularity for their versatility in building stable three-dimensional porous scaffolds with a broad range of internal microarchitectures to investigate numerically. This flexibility materialized in many scientific works currently focus on the optimization of the microstructure of scaffolds with different types of materials beyond the traditional biomedical metals, including bioceramics or polymers [16–18]. Although scaffolds have been implanted with reasonable success in the repair of small bone defects [19,20], their success with critical-sized ones remains limited [21,22].

After an orthopedic surgery of a critical-sized bone defect, continuous monitoring of the bone regeneration process is advisable, especially in the early weeks when bone infections or non-unions frequently appear. In this line, the progress of a bone healing process is typically monitored by plain film radiology and densitometric methods [23–25]. Beyond being accompanied by frequent exposure to radiation, the lack of continuity in this follow-up constantly results in qualitative conclusions and time lags in critical decision-making, including the need for reintervention or fixation removal. These delays occur since defect mineralization is primarily reflected in radiological images time after the real increase in bone callus stiffness [26,27]. Poor correlations of radiographic measurements with the tissue mechanical properties

have been consequently reported in the literature [27]. Thus, despite the development of scoring systems and indexes to evaluate and compare sets of radiographs [23,24], in practice, clinicians tend both to overestimate or underestimate a lower callus strength than it really is [28]. Other traditional *in vivo* manual methods, such as clinical symptoms (tenderness or pain while bearing weight) or the mobility review of the treated bone, trust the clinician's ability to interpret examinations and do not ensure the proper ossification in any case [29]. In the tissue engineering field, the opacity of its base material, its small pore sizes, and the generally high apparent stiffness of the structure could hinder performing a realistic analysis of the naïve tissue formation inside from radiological or manual assessments, especially in those critical first post-operative weeks. In recent decades, engineers have been developing alternatives that offer quantitative mechanical and structural parameters to predict bone regeneration indirectly. Loads through the fixation, interfragmentary displacements, strains, or acoustic emissions collected by means of a wide variety of sensors allow estimating the bone callus stiffness *in vivo* and mechanically comparing surgical protocols and bone regeneration approaches [5,30–35]. However, most instrumented systems can only be coupled into external fixators and require complex and expensive acquisition equipment not widely available today in the clinical routine. In vibrational measurements, the treated bone is also required to be subcutaneous [23,32].

As another option, gait analysis has been used to follow up on kinematics [36], electromyographic (EGM) [37], or load-bearing parameters [38–40] potentially modified by orthopedic surgery. During kinematics tests and EGM, measurements offer complete information regarding the position, orientation, and electrical activity of many body segments at the expense of the time-consuming tasks of attaching skin markers or electrodes to the patients' bodies [36,37]. Among all the potentially collectible biomechanical parameters, some authors have developed algorithms to optimize the best set that identifies significant differences in gait patterns under the specific conditions of each study [41,42]. By cons, the ground reaction force (*GRF*) measurement allows analyzing force-time curves collected by load-bearing platforms during the patient's stance phases. The simplicity and the possibility of directly examining these data *in vivo* with no complex data post-processing make it the most extrapolated-to-clinic method to study the dynamic evolution of the lower limbs after orthopedic treatments or surgeries. *GRF* has also been reported to reflect the recovery of apparent stiffness at the defect site until the full functional recovery in fracture healing and distraction osteogenesis processes [38–40]. It also makes it possible to compare the impact of surgery on bearing capacity between orthopedic treatments, as well as the speed of recovery in each case. In addition, clinical decisions based on *GRF* data are proven more reliable than those taken from kinematic or EGM parameters [43,44]. As far as the authors are concerned, gait analysis has not been tested in bone tissue engineering applications on lower limbs, and its usefulness in assessing the proper regeneration of bony defects treated with scaffolds remains unknown.

In this line, this work aims to verify the usefulness of gait analysis *in vivo* for assessing the bone regeneration of weight-bearing critical-sized defects treated with bioceramic scaffolds externally stabilized. The proposed monitoring methodology is clinicians-friendly through low-cost devices whose collected data do not require complex post-processing tasks. This work also delved into the potential correlation between the different analyzed gait parameters in the operated and non-operated limbs throughout the TE mineralization phase as a means to simplify gait analysis in real clinical scenarios. Besides, the recovery mode of the studied parameters, including data from the contralateral limb, will be compared with other equivalent orthopedic treatments on the same bone model (e.g., bone transport) to investigate the functional advantages of each regeneration process.

## Materials and methods

### Animals and bone model

TE experiments were performed *in vivo* on eight right-back ovine metatarsi of adult female Merino sheep (n = 8). A control group of 3 non-operated animals, randomly selected, were included in the study to contrast results with the experimental group directly (n = 3). Therefore, experiments were carried out in a total of 11 animals. Sample size was selected according to the standard deviation and statistical tests found in a previous study [39]. This study and its protocol were previously approved by the Animal Ethics of the University of Córdoba (Protocol Number: 2021PI/21) and guaranteed during surgical interventions and experimental phases the animal's welfare in strict accordance with the ARRIVE guidelines, the European (2010/63/UE) and national (RD 1201/2005) regulations on animal experimentation, thus avoiding stressful situations. A completed copy of the Full ARRIVE 2.0 Guidelines checklist is provided in the S1 Checklist of the Supporting Information. The selected mammalian model has numerous advantages. Sheep are docile, low-priced, and have a bone composition and body weight comparable to humans [45–47]. These similarities allow for easier extrapolation of conclusions to real clinical cases. The metatarsus was chosen as bone mode due to its thin layer of surrounding soft tissues, finding exclusively periosteal tissue on the lateral and medial sides. This anatomical advantage allowed a safer and more accessible tissue engineering surgery. The animals were obtained from a farm for research and were marked on the wool to avoid confounders. They were healthy, with an adequate vaccination and deworming protocol. Sheep with an adequate metatarsal length were selected for the study (> 12 cm). They were stabled 15 days before the start in the research center facilities for an adaptation period where all experiments were carried out. Their facilities were adapted to provide for their needs, including shade, comfortable lying areas, and secure fencing to keep them unstressed and safe. In the operated group, a 15-mm critical-size bone segment of right metatarsus was surgically replaced by a subject-specific bioceramic scaffold. For comparative reasons, this bone defect size was selected based on the size in other ovine bone regeneration experiments in the literature [4,5,39]. The defect was externally stabilized by a modular Ilizarov-type external fixator. This external solution minimizes body invasion compared to internal plates and splits. Hence, they reduce the soft tissue damage, leading to an early recovery of the animals' mobility and a lower impact on the analyzed gait parameters in the weeks immediately following the intervention. When the experimental part of the study concluded, the animals were euthanized by an overdose of sodium pentobarbital IV Euthasol®. The following subsections cover the design and manufacturing process of the scaffold, the steps of the orthopedic surgery, details about the *in vivo* gait experiments, and the radiological assessment. In the "Surgery procedure" subsection, all relevant steps taken to ameliorate animal suffering during the research are also detailed.

### Design and fabrication of the scaffold

The scaffold was designed with the subject-specific geometry of the bone fragment to be replaced. For this, computed tomography was performed on the right hindlimb of the sheep before surgery (voxel size 0.12 x 0.12 x 0.60 mm). A multiplanar hard tissue thresholding was applied to each metatarsal scan using the software InVesalius® (Renato Archer Information Technology Center, Amarais, Brazil), which enabled to generate a 3D bone reconstruction from the image stack, as shown in Fig 1A. Afterward, an intermediate 13-mm bone segment was sliced from the metatarsal geometry employing the solid modeling CAD software Space-Claim® (SpaceClaim Corporation, Concord, MA, USA), and its inner medullary cavity was filled. Two building modifications were also carried out on this initial geometry to improve

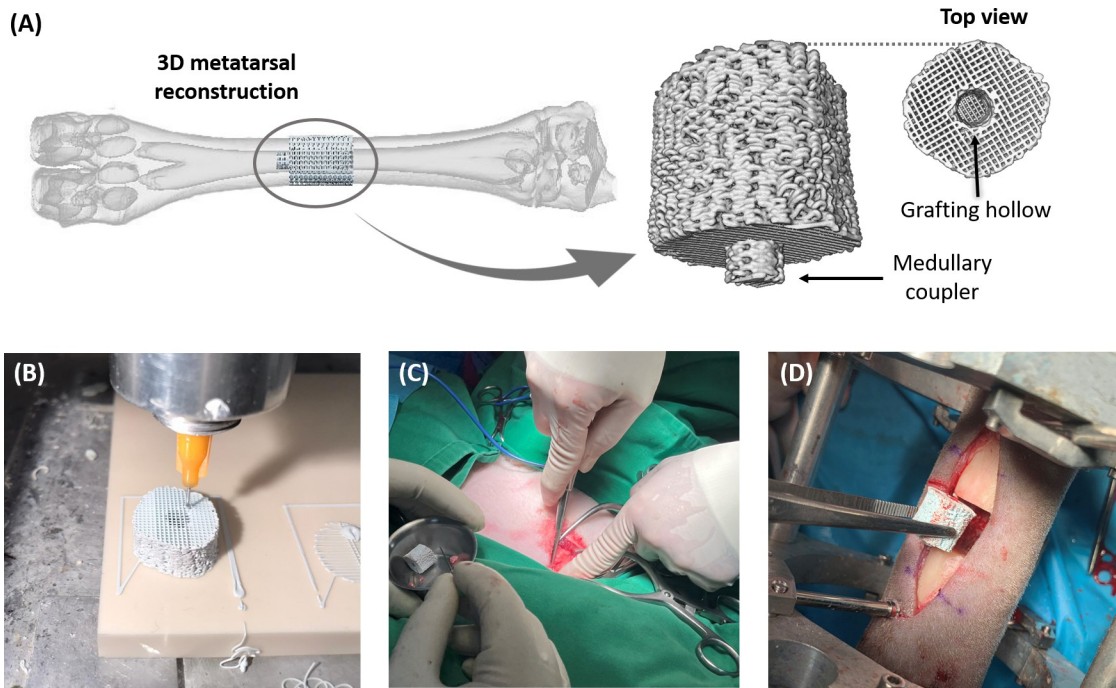

**Fig 1. Steps in the design of the bone tissue engineering surgery.** (A) Design of the bioceramic scaffold from the 3D geometry of the ovine metatarsus reconstructed by computed tomography scans. (B) 3D-printing of the hydroxyapatite structure using roboscasting. (C) Harvesting of the spongy grafting tissue from the lateral side of the contralateral humerus. (D) Implantation of the scaffold.

the scaffold stability in the defect and speed up the regenerative response (see Fig 1A). Firstly, a coupling cylinder (4 mm Ø; 2 mm length) was added over one end face to immobilize the structure *in vivo* in the distal bone marrow. This coupler would prevent the scaffold from rotating or moving during the animals' daily activity, which could interfere with the proper growth of the naïve tissue. At the other extreme, the solid was hollowed out by another cylinder (4 mm Ø; 10 mm length) for grafting, as further explained. A robocasting device (3-D Inks Still-water®, Tulsa, Oklahoma, USA), an extrusion-based 3D-printing technique, was selected. It worked by depositing a 45 vol% hydroxyapatite slurry forming a ceramic network of perpendicularly oriented layers of bars, as illustrated in Fig 1B. This material was chosen since bioceramics are proven to enhance naïve tissue growth by regulating osteoblast proliferation and differentiation while the structure is reabsorbed [48]. From previous works in the literature, the concentration of colloidal suspensions in the 3D-printing ink (45 vol%) was optimized to obtain the suitable viscoelastic properties for an effective deposition and assembly while ensuring the proper mechanical performance of the sintered structure *in vivo* [49–51]. An example of the final geometry of the patient-specific scaffold is provided in the S1 File of the Supporting Information. The printing nozzle diameter, the pore size, or the layer overlap were also numerically optimized in previous works to maximize cell diffusion and proliferation while ensuring the mechanical integrity of the structure under the ovine physiological loads [17]. The final microarchitecture had a porosity of 59.3%, a 560.8 μm pore size, and a specific surface area of 5768.9 m$^{-1}$ [17]. After drying at room temperature, the organic components of the implant were eliminated under heating at 400°C for 1 hour. They were finally sintered at 1300°C for 2 hours to compact the paste-like scaffold. The chemical sterilization of the structures was achieved using formaldehyde at 60°C and relative humidity of 80%.

## Surgical procedure

Before surgery, Amoxiciline 15 mg/kg Clamoxyl® IM and Meloxicam 0.2 mg/kg Metacam® IV were supplied to the animals to prevent infections and excessive inflammation. Surgery was carried out under general anesthesia induced by detomidine 20 μg/kg Detogesic® IV and morphine 0.2 mg/kg Morfina B.Braun® IM and was maintained with the inhalational anesthetic isoflurane IsoVet® 1–1.2% transported in 100% oxygen. The sheep's body temperature, blood pressure, oxygen saturation, expired fraction of carbon dioxide and electrocardiograms were constantly monitored during the intervention.

The animal was placed in right lateral decubitus for a medial approach. The right hindlimb was shaved, and its skin was aseptically prepared using chlorhexidine and an antiseptic alcohol-based solution. An 8-cm incision was then performed using an electrical scalp to expose, after a careful separation of the periosteal tissue, the underlying metatarsal segment to replace. Before osteotomizing, an Ilizarov-type external fixator was implanted in the metatarsus to keep the resulting unconnected bone fragments aligned. This stabilizing frame comprises two aluminum rigs interconnected by metallic bars and fixed to the bone through a total of six drilled 4-mm Ø Schanz pins. More information about the external fixation design and its mechanical properties is provided in Blázquez-Carmona et al. [52]. Two parallel transversal osteotomies 15 mm apart from each other were made in the intermediate part using an oscillating saw. As a result, a critical-size 15 mm defect was created in each animal. Adding bone morphogenetic proteins or grafted tissue to the scaffold is an extended strategy used in tissue engineering experiments to avoid a lack of spontaneous healing in critical-size defects [21,22]. In this line, cancellous bone autograft from the lateral side of the contralateral humerus head was immediately harvested through a 7-cm incision using Volkmann spoons, as shown in Fig 1C. This spongy tissue was inserted in the inner hole of the scaffold, working as a cell-seeding vehicle to accelerate osteogenesis. The scaffold was then implanted into the defect and fixed to the distal fragment by introducing the coupler into the bone marrow, as illustrated in Fig 1D. Since the scaffold's main body was 13 mm long, the remaining 2 mm proximal gap was filled with the remaining cancellous grafting tissue. After a quick radiological analysis to verify the scaffold implantation and the non-joint invasion of the fixator's pins, the operated sheep were recovered with oxygen under assistance. During the first five days after surgery, analgesia was also provided (meloxicam 0.1 mg/kg SC Metacam® and buprenorphine 0.03 mg/kg IM Bupaq®) according to ruminant pain scales. During the research, external fixators were periodically checked, and the skin-pin transitions were cleaned with chlorhexidine Desinclor®.

## Gait analysis

Gait tests consisted of making the sheep walk over a guided circular gangway containing a wireless pressure-sensitive platform Pasco PS-2141® (PASCO, Roseville, CA, USA), as can be seen in Fig 2A. The load platform (35 x 35 cm) continuously measures the dynamic vertical *GRF* in the animals' stance phases (sampling rate of 50 Hz), the largest component of the total *GRF* [53]. The platform was embedded inside the walking gait to isolate vibrational noises and enable measuring "steady-state" walking. Beyond the operated right ipsilateral hindlimb (*IH*), limb gait conditions were also analyzed to assess potential compensation mechanisms for loss of bearing capacity after tissue engineering intervention: the ipsilateral forelimb (*IF*), the contralateral hindlimb (*CH*), and the contralateral forelimb (*CF*). Fig 2B shows a scheme of these gait parameters on a control *GRF* curve. Prior to surgery, an acclimation period was also needed for each animal since they are naturally gregarious animals, thus being averse to walking in isolated settings [54]. After surgery and a latency and recovery period of 7 days, weekly measurement sessions were carried out by recording 7–10 stance phases per limb, thus

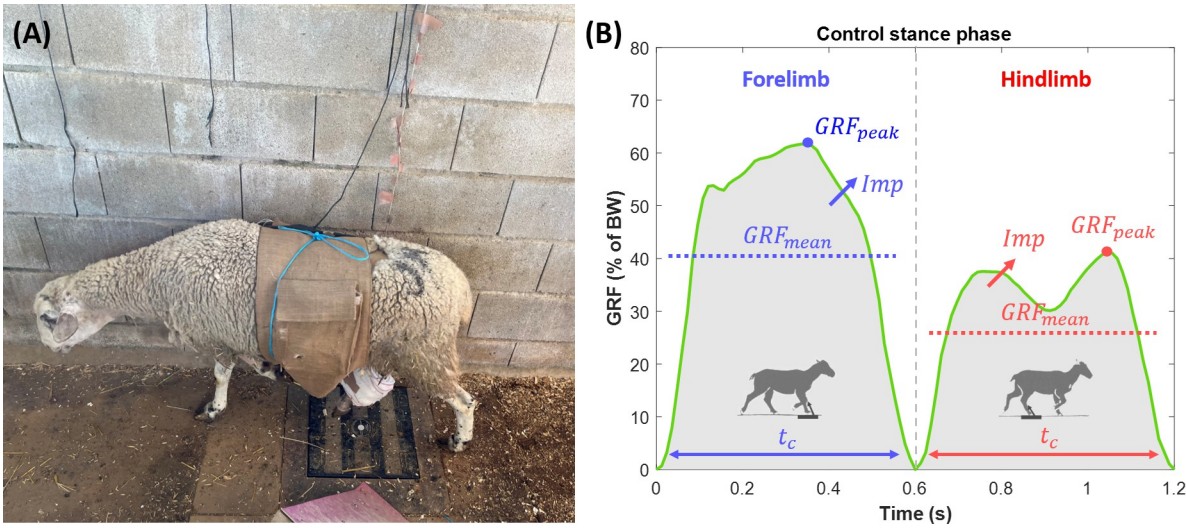

**Fig 2. Gait tests and analyzed parameters.** (A) One sheep of the study walking in the instrumented gangway containing the wireless load platform. (B) scheme of the measured gait parameters over a healthy ground reaction force curve (GRF) normalized by the bodyweight of the sheep.

analyzing the data as the daily average. Each gait phase for every limb was determined by analyzing the collected data, starting from the time-point when an increase in ground reaction force was recorded until the time-point at which it returned to 0 N. As inclusion criteria, acquired stance phases performed above or below amble speed (2–4 km/hr) were visually discarded. Moreover, data in which the animal interrupted its march at any point of the gangway were not included in the analysis either. During the experimental tests, a supervisor meticulously wrote down the specific time-points at which each limb executed a valid full stance phase on the load platform to further associate each GRF curve to their respective limb. The ground reaction force data from their recorded stance phases is provided in the S2 File of the Supporting Information.

As shown in Fig 2B, four main dynamic and kinematic parameters were analyzed in every collected stance phase per limb: the maximum ground reaction force ($GRF_{peak}$), the average ground reaction force in the stance phase ($GRF_{mean}$), the contact time with the ground ($t_c$), and the relative impulse ($Imp$) calculated as the area under the GRF normalized by $t_c$ to decouple the effect of gait speed. All the above parameters are standard in gait analysis experiments with load platforms [24,38,39]. They were additionally normalized by the body weight of each specimen ($BW$) to remove body-size dependence. Consequently, the weight of the animal was controlled throughout the entire experimentation period, especially during the first weeks after surgery when an appetite and a slight weight loss were commonly suffered by sheep. Furthermore, gait asymmetry between ipsilateral and contralateral limbs was assessed for each previous gait parameter according to the formulation employed in previous works [39,55]. This index for the parameter $x$ can be calculated using Eq 1.

$$Asymmetry(\%) = 100 \cdot \left| \frac{X_I - X_C}{0.5 \cdot (X_I + X_C)} \right| \tag{1}$$

where $X_I$ and $X_C$ are the daily mean values of each gait parameter for the ipsilateral and the contralateral limbs, respectively. The asymmetry index should be close to zero both for hind- and forelimbs in healthy specimens without any orthopedic pathology.

Statistical analysis was also performed using the software MATLAB® (The MathWorks Inc., Natick, MA, USA) to assess the potential dependency between all analyzed gait parameters of the four limbs. Precisely, the goodness-of-fit of linear regression models was measured through the coefficient of determination, R-squared and p-value. The coefficient of determination R-squared is traditionally interpreted as the percentage of variance in one variable predicted or explained by the other [56]. In this field, authors tend to interpret R-squared over 0.49 and 0.81 as a strong and very strong correlation, respectively [5,38,39]. The p-values for the coefficients indicate whether these previous relationships are or not statistically significant. Thus, a low p-value ($< 0.01$) would confirm that one gait parameter is significantly dependent of another.

## X-ray follow-up

In parallel with the gait analysis, the regeneration progress was verified through monthly x-rays of the operated metatarsal. These images were taken to clinically ensure the non-appearance of non-unions and to qualitatively associate changes detected in gait conditions with the level of ossification at the different regeneration stages.

## Results

### Gait parameters normalize over healing time

After a week of post-surgical recovery and latency, the sheep were able to walk appropriately on the instrumented gate. In general, the weight of the animals decreased from 57.90 ± 9.62 kg after surgery to 53.04 ± 6.34 kg before the sacrifice. Fig 3 compares the evolution of the *GRF*, normalized by the body weight (% *BW*), during gait tests at different time points of the consolidation phase, specifically weeks 3, 10, and 30 after surgery. For one experimental test in these weeks of one of the specimens, the curves show the means (dotted black lines) and standard deviations of the force of the 7–8 stance phases of all limbs: ipsilateral and contralateral forelimbs (*IF* and *CF*, blue curves) and hindlimbs (*IH* and *CH*, red curves). Note that the operated metatarsal in which the bioceramic scaffold replaced a critical-size bone fragment was the ipsilateral hindlimb (*IH*). The healthy control group data are also included (green curves). Comparing with these control curves, a remarkable alteration in gait conditions and limbs' loading capacity of the animal was found in the first weeks after surgery. First, the percentage of body weight carried by the operated *IH* was reduced by approximately 10%. Furthermore, similarly to humans, the control vertical force data of the sheep's hind limbs during a gait cycle has a very particular M-shape with four increasing and decreasing intervals. This behavior is fundamentally due to initial movements and accelerations of the animals' center of mass in the body weight distribution with the forelimbs, as well as the appearance of a propulsive action before push-off. This M-shape completely disappeared after surgery, reporting a single maximum *GRF* for punctual contact with the ground. In parallel, the rest of the limbs carried slightly more *GRF* during their stance phases at the beginning of the regeneration process (Fig 3, week 3): 11.2% of the *BW* above control in *IF*, 21.7% in *CF*, and 17.59% in *CH*. Nonetheless, the shape of the force curves was not drastically modified as the operated limb's. Fig 3 also shows that all limbs normalized the distribution of body weight and the bearing capacity of their limbs to healthy data throughout the consolidation weeks. For instance, the operated *IH* limb reported values similar to those of the control group at week 30, with 40% *BW* compared to 44% in non-operated sheep. However, the *GRF* curve of the *IH* limb did not regain the healthy M-shape in the medium term. Simultaneous to this recovery, the x-rays of Fig 3 show a gradual ossification of the critical-size bone defect and the gradual formation of a tissue bridge that consistently mineralized, connecting the original proximal and distal bone fragments.

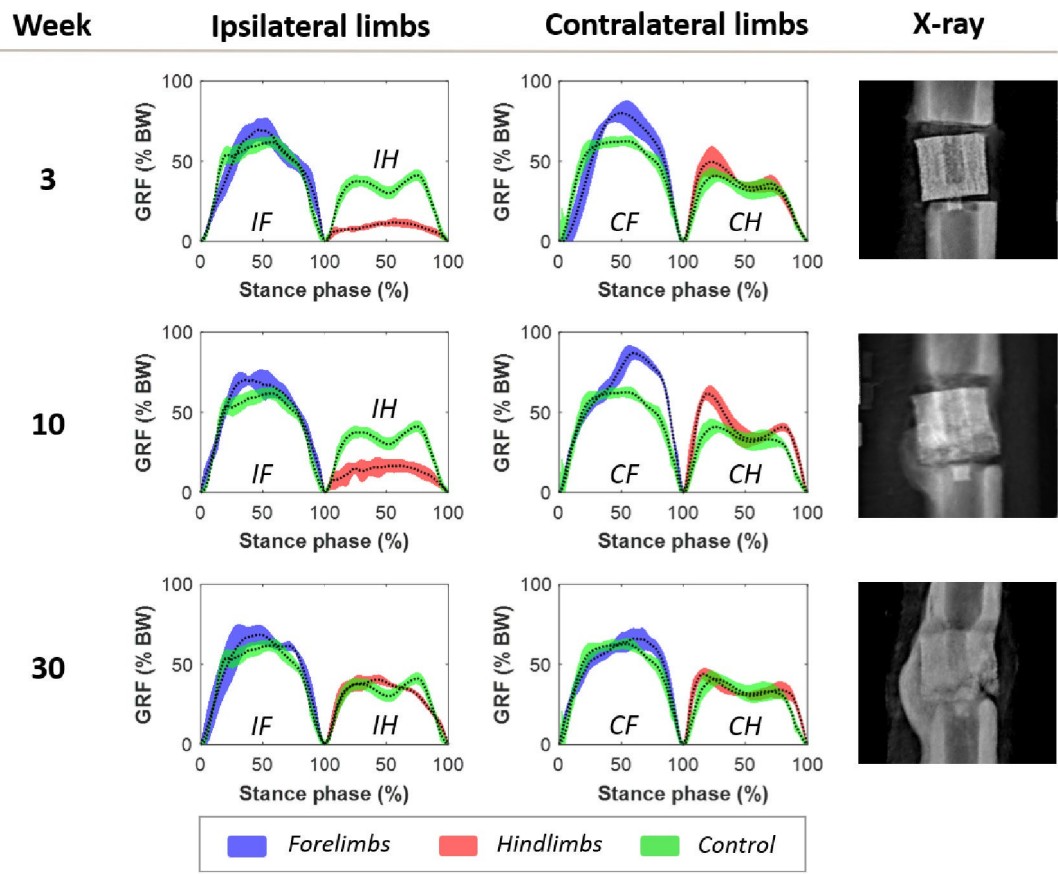

**Fig 3. Evolution of the ground reaction force curves over the weeks after surgery.** Ground reaction force curves (GRF) over the stance phases of the ipsilateral forelimb (IF), the operated right ipsilateral hindlimb (IH), the contralateral forelimb (CF), and the contralateral hindlimb (CH). Weeks after surgery and x-rays are included as a reference for the degree of ossification at said time-point. Control ground reaction force data is included for all limbs (green curves).

Similar trends were found when comparing the evolution of all the gait parameters analyzed for all the animals. The complete set of experimental data for these parameters is provided in the S3 File of the Supporting Information. For example, Fig 4A and 4B analyze the evolution of the $GRF_{peak}$ and $GRF_{mean}$ in the four limbs: operated $IH$ limb (red empty marker), $IF$ limb (blue empty marker), $CH$ limb (red filled marker), and $CF$ (blue filled marker). The means of the control data for the fore- and hindlimbs are also represented as dotted lines. Both parameters significantly reduced the $GRF$ levels after the TE surgery. Specifically, the replacement of the bone fragment by the bioceramic scaffold reduced the $GRF_{peak}$ to 15–23% of the $BW$ (57–163.46 N depending on the animal) and the $GRF_{mean}$ to 11–16% (39.52–95.13 N) when the control values were around 42% and 28% $BW$, respectively. These parameters took around 160 days to return to healthy values. The contralateral limb followed the inverse trend starting from 44–62% of the $BW$ for the $GRF_{peak}$ parameter (242.97–329.67 N depending on the animal) and from 30–39% of the $BW$ for the $GRF_{mean}$ (158.42–224.49 N). Likewise, the forelimbs also initially carried more body weight than their corresponding control during gait, 65% and 44% of the $BW$ (around 370 and 250 N) for $GRF_{peak}$ and $GRF_{mean}$, respectively. As reflected in Fig 4C, the regeneration process also impacted the $t_c$. Although the stepping time of a sheep is, on average, 0.58 s in hindlimbs and 0.52 s in forelimbs, depending on the specimen, the intervened limbs made contact with the ground between 0.25–0.47 s. In contrast, other limb's

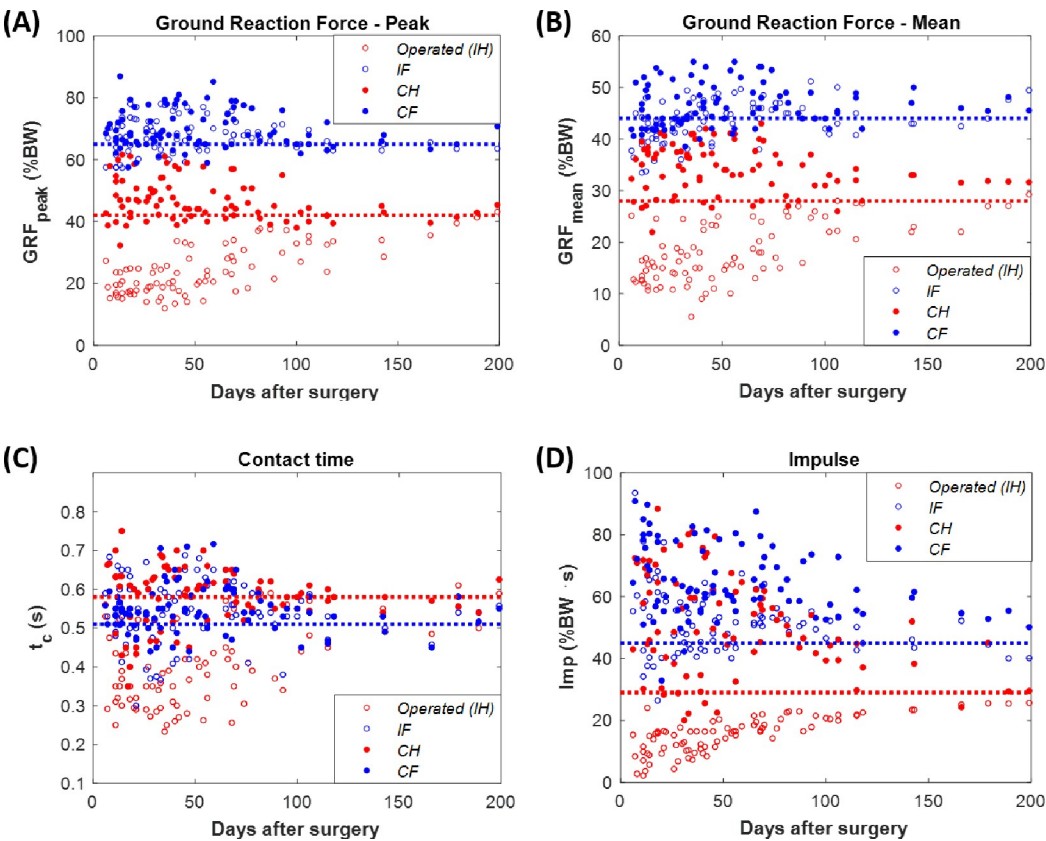

**Fig 4. Evolution in ovine gait-related parameters after a bone tissue engineering surgery.** (A) Maximum ground reaction force normalized by the animal's body weight (% BW). (B) Mean ground reaction force during the complete stance phase normalized by the bodyweight (% BW). (C) Contact time with the ground. (D) Gait impulse normalized by the body weight and the contact time with the ground (% BW·$t_c$). Control reference for hind- (red) and forelimbs (blue) is included as dotted lines.

stance phases lasted slightly longer, additional 0.05 s for each on average. As in the force, the change in this parameter was temporary, and regular gait conditions ended up being recovered after 200 days of consolidation. Lastly, a similar pattern was found with the impulse (*Imp*, Fig 4D). In this case, the operated limb showed an impulse between 0–20% *BW*·s after the operation when the hind control data was 29%. Preoperative impulse levels were not reached until day 140 of consolidation. Even though the control for forelimbs is 45% *BW*·s, both the rest of the hind- and forelimbs kinematically propelled the body in a very similar way, showing results with high dispersion, between 37–93%. Once again, all of them tended to a gradual recovery.

## Gait asymmetries are attenuated during the regeneration

On the other, Fig 5 evaluates the level of asymmetry between the hind- (red markers) and between the forelimbs (blue markers) for each of the previous parameters. The complete set of asymmetry data is provided in the S4 File of the Supporting Information. Both for the *GRF* parameters and for $t_c$, forelimbs showed a low asymmetry below 25%, while ranging mainly between 30 and 120% in hindlimbs. At the end of the experiments, this pathology was minimized to below 10% in all previous cases. As shown in Fig 5D, the impulse did report more significant asymmetries that even exceeded 150% in the hind limbs. Its normalization during the consolidation phase also did not drop below 13% at 200 days after surgery.

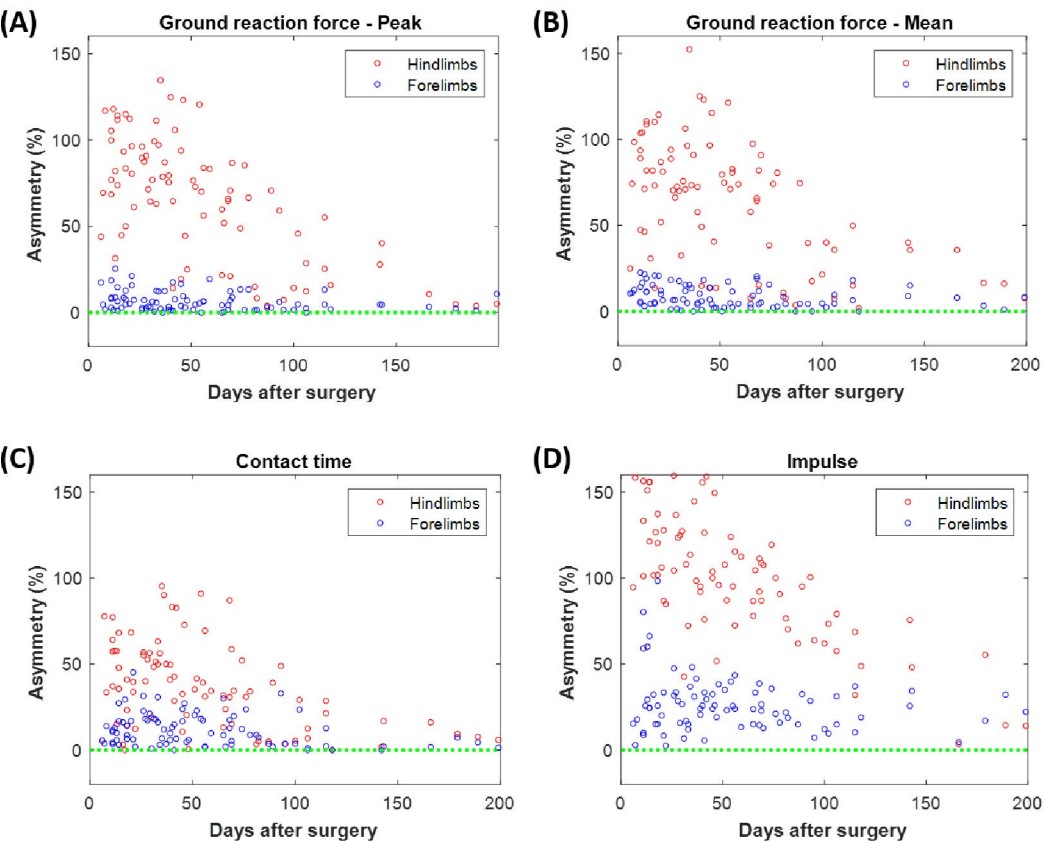

**Fig 5. Evolution of the level of asymmetry between the ipsilateral and contralateral hind- and forelimbs.** Asymmetry in the following parameters: (A) maximum ground reaction force; (B) mean ground reaction force; (C) contact time with the ground; (D) impulse. Data from hindlimbs are presented with red markers, and data from forelimbs in blue. Control reference is included as a green dotted line.

## Improvements in the gait of the different limbs are not independent

The possible correlation in recovering the previous parameters for all the animals' extremities was also analyzed. Fig 6 assesses the coefficients of determination after building linear regression models between them. Specifically, a table is shown with the individual R-squared for each pair of parameters, and those correlations in which the p-value is below 0.01 are indicated. Performing a global analysis, the gait parameters of the same limb seemed to correlate significantly well among themselves, except for the *CH*. The evolution of the *IH* parameters also described a close relationship with the regeneration time. Significant linear regressions were also found between ipsilateral limbs and between contralateral ones, reporting some R-squared above 0.5. As expected, the maximum and average *GRF* reported the best correlations, with R-squared generally above 0.70 and p-values < 0.01. In contrast, $t_c$ was the parameter that showed the most negligible significance with the rest.

## Discussion

This work demonstrated that gait analysis could potentially monitor the regeneration of critical-size defects treated by TE. Research in this field is primarily focused on optimizing the substitute's inner microarchitecture *in silico* [16,17] or validating its regenerative properties *in vivo* [21,22]. Most experiments resort to x-rays for a qualitative *in vivo* follow-up and mechanically and biologically characterize the tissue *ex vivo* through torsion tests, histology, or micro-

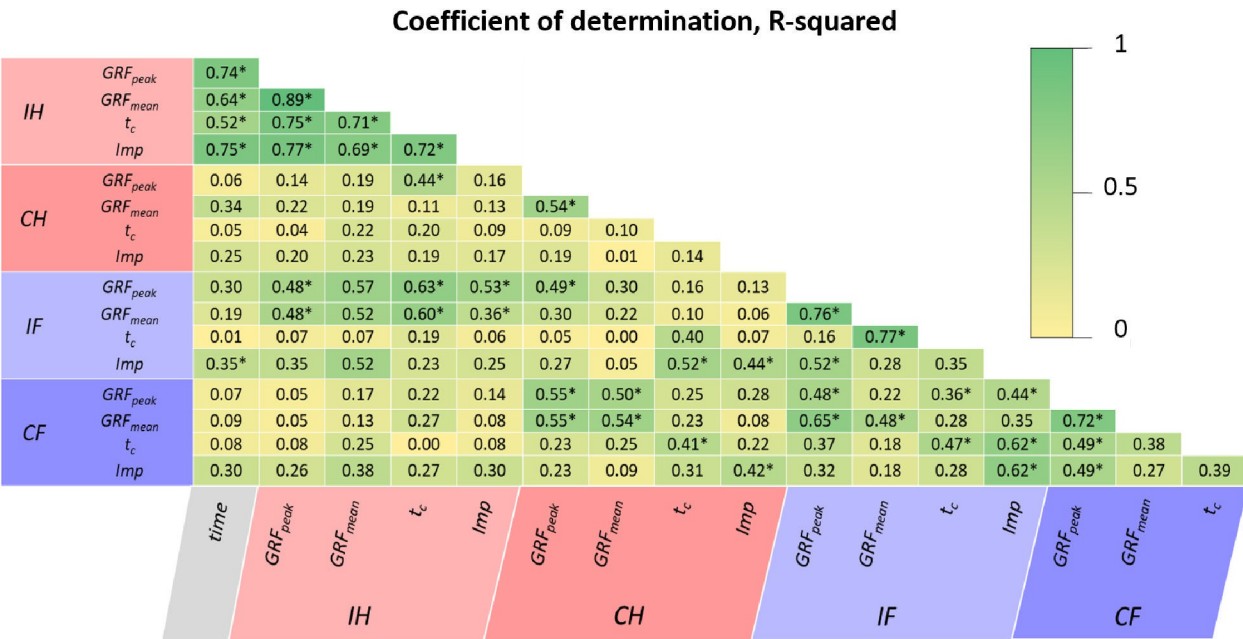

**Fig 6. Goodness-of-fit of linear regression models between gait-parameters and limbs.** Coefficient of determination, R-squared, in linear regression models was between consolidation time and gait parameters of all limbs: operated ipsilateral hindlimb (*IH*), contralateral hindlimb (*CH*), ipsilateral forelimb (*IF*), contralateral forelimb (*CF*). *p-value < 0.01.

CT [19–22]. As far as the authors know, there is no continuous quantitative monitoring of the regeneration throughout the TE consolidation phase in the literature. In this sense, gait analysis opens the door to more precise clinical control of these innovative treatments in traumatology. This methodology is also presented as a low-cost research tool that would make it possible to compare and optimize implants in their ability to recover the walking conditions of the patient after a TE surgery. Using animal bone models without excessive surrounding soft tissue, such as the ovine metatarsus, has the advantage of not influencing gait healing with possible soft tissue damage during the intraoperative period. Isolating the influence of these secondary lesions, minor improvements in the recovery of gait parameters during the first weeks after surgery are associated with problems in defect healing, thus diagnosing *in vivo* the appearance of non-unions [57].

Figs 3 and 4 show that despite anti-inflammatory treatments, TE surgery induced an expected lameness in the operated limb in all specimens. This condition is reflected in all main studied parameters: $GRF_{peak}$, $GRF_{mean}$, $t_c$, and *Imp*. As mentioned above, there is a gap in the literature in the application of this analysis in TE experiments. However, it is possible to compare our results with other *in vivo* studies of other pathologies or regeneration processes. The most direct comparison could be carried out with the bone transport study of Mora-Macías et al. [39]. They regenerated a 15-mm critical-size ovine metatarsal defect stabilized with a similar external fixation throughout distraction osteogenesis. In this bone regeneration process, the reduced mobility of the operated limb was significantly less pronounced. For example, the maximum ground reaction force ($GRF_{peak}$) was barely below 40% *BW* after surgery [39], while in TE, it fell to approximately 20% *BW* (Fig 4A). Comparing the $GRF_{mean}$ also illustrates these different mobility losses: around 25% *BW* for bone transport versus the 10–20% *BW* range reported by the animals of the current study (Fig 4B). The same results are obtained by analyzing the *Imp* parameter, which barely dropped below 25% *BW·s* in distraction osteogenesis. These bone transport results align with the gait analysis performed by Fischer et al. [58] in

dogs with general lameness. Under the same distraction protocol and fixation, Blázquez-Carmona et al. [38] also assessed gait parameters in sheep after inducing a 15-mm metatarsal callus by applying a gradual limb lengthening. In this regeneration process, the lameness was even more remarkable, and the vertical ground reaction forces collected were only slightly higher than 10% *BW* after distraction [38]. In all these regeneration processes, all the parameters were restored up to the full functional status but at different healing speeds. Assuming a linear evolution, bioceramic scaffolds recovered ground reaction forces at a rate of 3.6% *BW* per month, while bone lengthening normalizes this physical dysfunction at 4.9% *BW* [38]. This rate is lower in bone transport due to its less significant loss of initial bearing capacity, but they still reported values similar to healthy ones at only 20 weeks of consolidation [39]. The explanation of this difference between processes could be found in the principles of repair that it applies. Callotasis gradually separates with a specific distraction rate, 1 mm/day in the previous studies, bone fragments while ensuring their connection through a bone callus in continuous formation. This connective soft tissue serves as a template for subsequent ossification. Thus, TE must deal directly with repairing the critical-size defect without any initial connection between proximal and distal bone fragments. This could induce greater pain or a lack of confidence in the specimen's injured limb. Despite the presence of the bony callus, the more significant initial immobility of the limb treated with bone lengthening must be attributed to damage to the surrounding soft tissues unable to adapt to the induced elongation [5–7].

In our experiments, gait patterns of the non-operated limbs also changed due to involuntary compensation and adaptation mechanisms (see Figs 3 and 4). They logically tend to temporarily carry more % of the animal's *BW* during their stance phases and to lengthen their contact with the ground for several milliseconds. Nonetheless, the most precise way to analyze their behavior is through asymmetry factors, as presented in Fig 5. These indices directly compare the patient's injured and untreated limbs, avoiding the influence of intrinsic factors of each individual specimen, such as its age, skeletal maturity, or usual amble speed [18,59]. In general, the asymmetry has been proven to be high shortly after fracture cases and decreases throughout the healing process [57]. For example, Larsen et al. [60] measured the asymmetry between human non-injured legs and their contralateral fractured at the tibial shafts. The single support time was 12.8% shorter in the treated limb and the asymmetry normalized after 12 months. Nevertheless, again in severe human tibial fractures, Patterson et al. [61] demonstrated how the asymmetry in the duration of the stance phases, shorter in the reconstructed leg, was maintained even after several years of follow-up. In quadrupedal animal models, Mora-Macías et al. [39] observed an asymmetry of 40% of the healthy gait patterns after a bone transport process, which was reduced to 7–8% after 60 weeks. In line with the differences found in the previous parameters, TE seems to cause a more pronounced asymmetry between the operated limb and its contralateral one, between 50–100% and even above 100% in the case of *Imp* parameter (see Fig 5). However, this pathology seems to be corrected faster than in previous references, reaching control values in just 200 days. Thus, once a hard tissue bridge has been formed through the scaffold, as shown in Fig 3 (weeks 10 and 30), the animals may rapidly regain confidence in their operated leg, and regular walking mechanisms could be restored at great speed. In the evolution of the asymmetry in the untreated forelimbs, no significant differences are found between both regeneration processes.

As far as the authors are concerned, no previous study has examined the potential correlation between different gait parameters. The most comparable analysis might be the significant correlations found by Jansen et al. [62] between gait patterns after human pilon fractures (loading and time-force integral) and several scoring systems for the patient's general mobility (e.g., visual analog scale, Phillips, or the AOFAS's Ankle-Hindfoot Rating System). As reflected in Fig 6, we demonstrated a close relationship in the recovery of temporal ($t_c$) and mechanical

gait parameters ($GRF_{peak}$, $GRF_{mean}$ and $Imp$): R-squared $> 0.5$; p-value $< 0.01$. Thus, the gradual recovery of bearing capacity in the operated limb as the callus ossifies entailed a longer contact time with the ground in its stance phases. This conclusion could greatly simplify the gait analysis to a single parameter that ensures the non-appearance of non-unions, thus facilitating the inclusion of this methodology in the daily clinical routine. Significant correlations between contralateral and ipsilateral limbs were also reported. This could indicate that the aforementioned gait compensation mechanisms would proportionally alter the mobility of both sides of the body and their normalization would not be independent. In quadruped animals, this factor could also be essential to analyze in other traumatic disorders or regeneration processes to monitor potential asymmetric recoveries between limbs on each side of the body or enduring disorders in the general mobility of the specimen. Previous asymmetric indices do not provide this information since they compare hind and forelegs with similar healthy gait patterns.

## Conclusions

In conclusion, gait analysis effectively monitors the regeneration of critical defects treated by TE. This study proves that combining bioceramic scaffolds with external fixation successfully normalized gait patterns and mobility symptoms. However, despite the greater surgical simplicity that the direct replacement of a fragment implies, gait recovery is slower than in other bone regeneration processes, such as bone transport. A force platform is demonstrated to be a useful low-cost tool in the field of traumatology and tissue engineering. Firstly, it allows us to compare scaffolds and implants of different biomaterials, microarchitectures, or enhancing bone morphogenetic protein from a new approach. Secondly, it offers indirect, non-invasive, continuous and quantitative *in vivo* monitoring of the regeneration process without complex biomechanical and kinematic knowledge required, easily transferable to the clinic.

## Supporting information

**S1 Checklist. Full ARRIVE 2.0 Guidelines checklist.**
(PDF)

**S1 File. Outer geometry of a patient-specific scaffold.** OBJ of the geometry of the scaffold designed from CT scans of an intermediate 13-mm ovine metatarsal bone fragment. The specific modifications made to the scaffold for stability and regenerative response are included.
(OBJ)

**S2 File. Control data of the stance phases.** Ground reaction force data from the recorded stance phases in the non-operated control sheep of the study (n = 3).
(XLSX)

**S3 File. Experimental data of the analyzed gait parameters.** Complete set of all experimental data for all analyzed gait parameters ($GRF_{peak}$, $GRF_{mean}$, $t$ and $Imp$) in all animals in the study
(XLSX)

**S4 File. Asymmetries between hind- and forelimbs.** Complete set of all data asymmetries in the analyzed gait parameters ($GRF_{peak}$, $GRF_{mean}$, $t$ and $Imp$) calculated between the hind- and forelimbs.
(XLSX)

## Author Contributions

**Conceptualization:** Juan Mora-Macías, Esther Reina-Romo.

**Data curation:** Pablo Blázquez-Carmona.

**Formal analysis:** Pablo Blázquez-Carmona.

**Funding acquisition:** Juan Mora-Macías, Esther Reina-Romo.

**Investigation:** Pablo Blázquez-Carmona.

**Methodology:** Pablo Blázquez-Carmona, Juan Morgaz, María del Mar Granados.

**Project administration:** Esther Reina-Romo.

**Supervision:** Juan Mora-Macías, Jaime Domínguez, Esther Reina-Romo.

**Visualization:** Pablo Blázquez-Carmona.

**Writing – original draft:** Pablo Blázquez-Carmona.

**Writing – review & editing:** Juan Mora-Macías, Juan Morgaz, Jaime Domínguez, Esther Reina-Romo.

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
