## [Decision Letter · Decision Letter 0]

22 Nov 2023

PONE-D-23-29572Gait analysis: an effective tool to mechanically monitor the bone regeneration of critical-sized defects in tissue engineering applicationsPLOS ONE

Dear Dr. Blázquez-Carmona,

Thank you for submitting your manuscript to PLOS ONE. After careful consideration, we feel that it has merit but does not fully meet PLOS ONE’s publication criteria as it currently stands. Therefore, we invite you to submit a revised version of the manuscript that addresses the points raised during the review process.

We look forward to receiving your revised manuscript.

Kind regards,

Yaodong Gu

Academic Editor

PLOS ONE

Journal Requirements:

2. As part of your revision, please complete and submit a copy of the Full ARRIVE 2.0 Guidelines checklist, a document that aims to improve experimental reporting and reproducibility of animal studies for purposes of post-publication data analysis and reproducibility: https://arriveguidelines.org/sites/arrive/files/documents/Author%20Checklist%20-%20Full.pdf Please include your completed checklist as a Supporting Information file. Note that if your paper is accepted for publication, this checklist will be published as part of your article.

"This project was supported by the Ministerio de Ciencia e Innovación (Government of Spain) through the grant number PID2020-113790RB-I00 awarded to JMM and ERR. https://www.aei.gob.es/convocatorias/buscador-convocatorias/proyectos-idi-2020-modalidades-retos-investigacion-generacion "

"This publication is part of the R+D project PID2020-113790RB-I00, financed by MCIN/AEI/10.13039/501100011033.'

"This project was supported by the Ministerio de Ciencia e Innovación (Government of Spain) through the grant number PID2020-113790RB-I00 awarded to JMM and ERR. https://www.aei.gob.es/convocatorias/buscador-convocatorias/proyectos-idi-2020-modalidades-retos-investigacion-generacion "

Additional Editor Comments:

The authors need to feedback some small questions to the reviewers.

Reviewers' comments:

Reviewer's Responses to Questions

**Comments to the Author**

1. Is the manuscript technically sound, and do the data support the conclusions?

Reviewer #1: Yes

Reviewer #2: Partly

2. Has the statistical analysis been performed appropriately and rigorously? 

Reviewer #1: Yes

Reviewer #2: Yes

3. Have the authors made all data underlying the findings in their manuscript fully available?

Reviewer #1: Yes

Reviewer #2: No

4. Is the manuscript presented in an intelligible fashion and written in standard English?

Reviewer #1: Yes

Reviewer #2: Yes

5. Review Comments to the Author

Reviewer #1: This work evaluates the potential clinical use of gait analysis for the mechanical assessment of a tissue engineering regeneration as an alternative to the traditional and hardly conclusive manual or radiological follow-up. Nevertheless, there are areas that could benefit from refinement in order to enhance the overall clarity and precision of the manuscript.

Introduction:

Your introduction nicely emphasizes the role of tissue engineering in addressing critical-size bone defects. However, it would be great if you could clearly spell out how your study brings something new to the table in this field.

Line96-112

The gait analysis is well-established. However, you could briefly highlight why gait analysis is a promising avenue for monitoring bone regeneration, especially in weight-bearing critical-sized defects.

Materials and Methods:

The reasoning behind selecting sheep as your animal model and a 15mm metatarsal defect as a critical-size bone defect needs more thorough explanation. Additionally, delving into the specifics of why you chose the particular bioceramic scaffold and external fixator would significantly enhance the manuscript. Providing more 3D-printing details on these decisions will enrich the comprehensibility of your research.

Line152-154

Clarify the rationale behind the specific modifications made to the scaffold design for stability and regenerative response. Additionally, specify the reasons for selecting a 45 vol% hydroxyapatite slurry for the 3D-printing process and the significance of the resulting microarchitecture parameters.

Line202

While the gait analysis section is detailed, consider briefly explaining the rationale behind choosing the specific gait parameters analyzed (GRF peak, GRF mean, tc, Imp) and their relevance to assessing the bone regeneration process.

Line248

Elaborate on the frequency and specific aspects evaluated in the periodic X-rays. How did the X-ray observations correlate with the gait analysis findings, if at all?

Results

Enhance the clarity of the results presentation by including specific numerical values for changes in gait parameters. This will assist readers in comprehending the magnitude of the observed alterations more effectively.

Reviewer #2: This manuscript entitled “Gait analysis: an effective tool to mechanically monitor the bone regeneration of critical-sized defects in tissue engineering applications,” evaluates the potential clinical use of gait analysis for the mechanical assessment of tissue engineering regeneration. The authors demonstrated that gait analysis is an effective, low-cost tool by conducting gait tests after metatarsal replacement in sheep. Here below are specific comments:

1. In the Introduction, the logic needs to be strengthened regarding the role of eliciting gait analysis in evaluating recovery after scaffolding treatment.

2. Lines 96-98: “As another option, gait analysis has been used to follow up on kinematics…”, the reviewer agrees with this as gait analysis has become increasingly important in a variety of clinical assessments. perhaps the authors can refer to below research literature to further verify the opinion: (https://doi.org/10.1016/j.gaitpost.2023.10.019;
https://doi.org/10.1038/s41598-022-07054-1 ).

3. Why were sheep chosen as subjects?

4. How is the gait phase determined? The gait of a sheep is different from that of a human, and this affects the trend of the ground reaction forces captured on the force plates.

5. Line 237: Please refer to the relevant studies.

6. Lines 244-245: For the description of linear regression models, please add the more detailed information (https://doi.org/10.1016/j.cmpb.2023.107761). For example, how to interpret the correlation coefficients, such as what is a high correlation and what is a low correlation.

7. The results section is best broken down into smaller points to expand on, which will make the overall logic and content clearer.

8. The clarity of the pictures could be improved.

6. PLOS authors have the option to publish the peer review history of their article (what does this mean?). If published, this will include your full peer review and any attached files.

Reviewer #1: No

Reviewer #2: No

---

## [Author Response · Author response to Decision Letter 0]

1 Dec 2023

Responded in the "Response to Reviewers" file

---

## [Decision Letter · Decision Letter 1]

13 Dec 2023

Gait analysis: an effective tool to mechanically monitor the bone regeneration of critical-sized defects in tissue engineering applications

PONE-D-23-29572R1

Dear Dr. Blázquez-Carmona,

We’re pleased to inform you that your manuscript has been judged scientifically suitable for publication and will be formally accepted for publication once it meets all outstanding technical requirements.

Kind regards,

Yaodong Gu

Academic Editor

PLOS ONE

Additional Editor Comments (optional):

Well done!

Reviewers' comments:

Reviewer's Responses to Questions

**Comments to the Author**

1. If the authors have adequately addressed your comments raised in a previous round of review and you feel that this manuscript is now acceptable for publication, you may indicate that here to bypass the “Comments to the Author” section, enter your conflict of interest statement in the “Confidential to Editor” section, and submit your "Accept" recommendation.

Reviewer #1: All comments have been addressed

Reviewer #2: All comments have been addressed

2. Is the manuscript technically sound, and do the data support the conclusions?

Reviewer #1: Yes

Reviewer #2: Yes

3. Has the statistical analysis been performed appropriately and rigorously? 

Reviewer #1: Yes

Reviewer #2: Yes

4. Have the authors made all data underlying the findings in their manuscript fully available?

Reviewer #1: Yes

Reviewer #2: Yes

5. Is the manuscript presented in an intelligible fashion and written in standard English?

Reviewer #1: Yes

Reviewer #2: Yes

6. Review Comments to the Author

Reviewer #1: Thank you very much for submitting your article. After careful review, I'm pleased to inform you that your article has been thoroughly examined and found to be without any issues. Thank you for your hard work and excellent contribution

Reviewer #2: (No Response)

7. PLOS authors have the option to publish the peer review history of their article (what does this mean?). If published, this will include your full peer review and any attached files.

Reviewer #1: No

Reviewer #2: No

---

## [Editor Report · Acceptance letter]

19 Dec 2023

PONE-D-23-29572R1 

PLOS ONE

Dear Dr. Blázquez-Carmona, 

I'm pleased to inform you that your manuscript has been deemed suitable for publication in PLOS ONE. Congratulations! Your manuscript is now being handed over to our production team.

Kind regards, 

on behalf of

Professor Yaodong Gu 

Academic Editor

PLOS ONE